# Reliable Prediction of Post-Operative Complications’ Rate Using the G8 Screening Tool: A Prospective Study on Elderly Patients Undergoing Surgery for Kidney Cancer

**DOI:** 10.3390/jcm11133785

**Published:** 2022-06-30

**Authors:** Fabio Traunero, Francesco Claps, Tommaso Silvestri, Maria Carmen Mir, Luca Ongaro, Michele Rizzo, Andrea Piasentin, Giovanni Liguori, Francesca Vedovo, Antonio Celia, Carlo Trombetta, Nicola Pavan

**Affiliations:** 1Urology Clinic, Department of Medical, Surgical and Health Sciences, Cattinara Hospital, University of Trieste, Strada di Fiume, 447, 34149 Trieste, Italy; claps.francesco@gmail.com (F.C.); ongarluc@gmail.com (L.O.); mik.rizzo@gmail.com (M.R.); andrea.piasentin23@gmail.com (A.P.); gioliguori33@gmail.com (G.L.); francesca.vedovo@gmail.com (F.V.); trombcar@units.it (C.T.); nicpavan@gmail.com (N.P.); 2Department of Urology, San Bassiano Hospital, 36061 Bassano del Grappa, Italy; tommaso.silve@gmail.com (T.S.); antoniocel70@yahoo.it (A.C.); 3Department of Urology, Valencian Oncology Institute Foundation, FIVO, 46009 Valencia, Spain; mirmare@yahoo.es

**Keywords:** G8, elderly, preoperative assessment, frailty, geriatric assessment, urological surgery, kidney cancer

## Abstract

In the last years the incidence of renal neoplasms has been steadily increasing, along with the average age of patients at the time of diagnosis. Surgical management for localized disease is becoming more challenging because of patients’ frailty. We conducted a multi-center prospective study to evaluate the role of the G8 as a screening tool in the assessment of intra and post-operative complications of elderly patients (≥70 y.o.) undergoing surgery for kidney cancer. A total of 162 patients were prospectively enrolled between January 2015 to January 2019 and divided into two study groups (frail vs. not-frail) according to their geriatric risk profile based on G8 score. Several factors (i.e., age, CCI, ASA score, preoperative anemia, RENAL score, surgical procedures, and techniques) were analyzed to identify whether any of them would configure as a statistically significant predictor of surgical complications. According to the G8 Score, 90 patients were included in the frail group. A total of 52 frail patients vs. 4 non-frail patients developed a postoperative complication of any kind (*p* < 0.001). Of these, 11 were major complications and all occurred in the frail group. Our results suggest that the G8 screening tool is an effective and useful instrument to predict the risk of overall complications in elderly patients prior to renal surgery.

## 1. Introduction

Kidney cancer is very common, especially among elderly patients in western countries, and is often incidentally diagnosed during abdominal radiological investigations performed for other purposes. Renal Cell Carcinoma (RCC) is a very heterogeneous disease comprising both small renal masses (SRMs) incidentally discovered at the time of abdominal imaging and aggressive disease with de novo metastatic spread in which surgery is still an option in selected candidates [1,2,3].

Both the aging of the population and the incidence of RCC are growing, making the issue of oncological management in the elderly a real challenge, especially in the case of localized kidney neoplasms [4,5,6]. It seems clear that age is one of the main contributing factors to the functional decline of an individual, but we need to consider other factors to plan the right therapeutic strategy. The elderly have always been considered to be at risk and often do not undergo major surgery for fear of complications or poor subsequent recovery, even though they could have benefited from it [7]. Tests such as Comprehensive Geriatric Assessment (CGA) have been introduced to evaluate the real physical and psychological health conditions of the patients [8].

Although CGA has the advantage of being the most complete evaluation tool available to identify the medical condition of the elderly patient, due to its complexity it does not appear a suitable instrument in clinical practice, especially in conditions where a rapid evaluation must be made. For this strictly practical reason, geriatric scores have the aim of “filtering” a patient who needs a complete CGA and obtaining a global view of the patient’s health [9]. 

Regarding surgical treatment for elderly patients undergoing surgery for kidney cancer, the EAU (European Association of Urology) does not identify any specific guidelines, leaving the decision on which treatment to perform to the single urologist, based on the patient’s health condition. Thus, the clinical need to introduce standardized methods that allow an objective evaluation of patients emerges. Our study aims to analyze the role of the G8 score in predicting intra and postoperative complications in cases going to be submitted to renal surgery.

## 2. Materials and Methods

### 2.1. Patients Characteristics

Demographic, clinicopathological features, and postoperative outcomes data of elderly patients who underwent kidney surgery for clinical-localized renal masses were prospectively collected. All procedures were performed at two academic centers between January 2015 to January 2019. Obligated data-sharing agreement was decided upon before initiation. Data were collected with the approval of each institution’s Ethical Committee, and the study was conducted following the principles outlined in the Declaration of Helsinki. All subjects gave informed consent to participate in the study. The analysis included 162 elderly patients. Definition of elderly was age equal or superior to 70 years, regardless of other parameters [10]. Patients who were candidates for other ablative treatment (Radiofrequency, Cryoablation) or who had any previously active surveillance and/or other ablative treatment were excluded. The analysis included 162 elderly patients.

Variables collected were age, gender, Eastern Cooperative Oncology Group (ECOG) [11], Charlson Comorbidity Index (CCI) [12], Body Mass Index (BMI), American Society of Anesthesiologists Classification (ASA) score [13], preoperative anemia and estimated glomerular filtration rate (eGFR) according to the Chronic Kidney Disease Epidemiology Collaboration (CKD-EPI) formula, clinical size of the renal masses, R.E.N.A.L. score [14], clinical and pathological T stage [15], Fuhrman grade [16]. Surgical variables included approach and technique, intraoperative blood loss, intraoperative complications’ rate, and the presence of lymph node dissection (LND). For partial nephrectomy, ischemia features were also evaluated. Intraoperative complications were recorded by the primary investigator as a none or yes according to the event. Recently, an intraoperative complication score was proposed, but it was not available at the time of the study was conducted [17]. Perioperative surgical complications were reported following the Clavien Dindo criteria [18]. Length of hospital stay (LOS) and 30-days readmission were recorded. Prior to surgery, all patients underwent standard laboratory tests, physical examination, computed tomography (CT) of the abdomen/pelvis and chest for a complete staging.

Further radiological investigations were conducted if patients presented symptoms. Surgical procedures were performed either with a minimally invasive or open approach. All the specimens were analyzed by two dedicated genitourinary (GU) pathologists at each center and were staged based on the Tumor Nodes Metastasis (TNM) classification system (2017 classification, 8th edition). 

### 2.2. The G8 Screening Tool

The G8 screening tool includes seven items from the Mini Nutritional Assessment (MNA) and an age-related item (<80, 80–85, or >85 years). Total score ranges from 0 to 17. G8 result is considered abnormal if the score is ≤14, indicating a geriatric risk profile [19]. The time needed to complete the questionnaire is between 2 and 3 min. G8 items are listed in Appendix A. 

### 2.3. Endpoints

Th role of G8 score in predicting intra and postoperative complications was set as the primary endpoint of our current analysis. Intra-operative complications evaluated were bleeding with the need for transfusion, injuries to organs adjacent to the kidney or retroperitoneal vessels, superficial wound infections or deep infections, pneumothorax. All postoperative complications were recorded and classified according to the Clavien-Dindo classification.

### 2.4. Statistical Analysis

The 14-points cut-off value set by the literature for the G8 screening tool was tested using the maximum Youden index [20]. Patients were divided into two groups according to the geriatric risk profile based on the G8 score: patients with a G8 lower than the cut-off value were defined as “frail”, while the remaining patients (G8 over the cut-off value) were assigned to the “not-frail” group. Descriptive analysis included frequencies and proportions for categorical variables. Medians and interquartile range (IQR) were reported for continuous coded variables. The Kruskal Wallis or Mann–Whitney U test was used for comparison of the continuous data and the Chi-square or Fisher’s exact test for categorical data. All tests were two-sided with a level of significance set at *p* < 0.05. The efficacy of G8 was analyzed using the receiver operating characteristic (ROC) curve and the area under the curve (AUC). Univariate and multivariate binomial logistic regression models were used to assess the Odds Ratio (OR) with 95% CI testing the ability of G8-based groups to predict perioperative morbidity after adjusting for all common preoperative covariates. Statistical analyses were performed using RStudio Version 1.2.5001 (RStudio: Integrated Development for R. RStudio, Inc., Boston, MA, USA, URL http://www.rstudio.com/, accessed on 1 January 2020).

## 3. Results

### Descriptive Analysis

Table 1 is resuming the baseline, surgical characteristics, and clinicopathological features for the cohort of 162 patients with renal mass treated with radical or partial nephrectomy according to G8 Screening Tool status.

A total of 162 patients were enrolled in the study, of which 91 patients were categorized as ‘frail’ (56.2%) and 71 (43.8%) not-frail. Of the overall population, 92 (56.8%) were men and 70 (43.2%) were women. In the group of frail patients, the number of women and men was similar, while in the group of non-frail patients women were about twice as numerous. The median age at surgery was 76 years (range 70–91 years), median BMI was 24.9, and mean ECOG score of 1.54 (range 0–3). A total of 41 patients, identified as frail by the G8 score, versus 2 not-frail patients developed intraoperative complications (*p* < 0.0001). Comprehensively 55 postoperative complications occurred (34%), of which 11 in grade Clavien >3 (6.7%); of these, 4 required a new hospitalization within 30 days. The mean length of the first hospitalization was 10 days (SD ± 7.57). A total of 14 patients reported a benign histotype on final histopathological analysis, with no statistically significant differences between the fragile and non-fragile groups.

The ROC analysis for the G8 score showed that the AUC value predicting the overall complications’ rate was 0.91 (95% CI, 0.86–0.96) (Figure 1). According to the maximum Youden index value, the cut-off for G8 was set at 14. Therefore, 91 (56.2%) patients were classified into the “frail” group, whereas the remaining 71 (43.8%) patients were classified into the “not-frail” group.

We performed a Univariate and Multivariate Logistic Regression Analysis of factors significantly associated with perioperative complications according to Clavien-Dindo Classification (Table 2). None of the single predictors of complications considered in the analysis like age, CCI, ASA score, preoperative anemia, RENAL score, surgical procedures, and techniques showed a statistically significant association in predicting possible complications as opposed to the G8 score (*p* = 0.03). All complications were considered, as the number of major complications was too few (CCI >3 = 11) to perform a statistically significant analysis. Both univariate and multivariate analysis confirmed, with statistically significant *p* values (*p* < 0.001 for univariate and *p* = 0.03 for multivariate), that the G8 is a reliable instrument to predict postoperative complications. CCI, on the other hand, proved statistically significant only as regards the prediction of complications in patients with scores higher than two (*p* = 0.04).

## 4. Discussion

The global increasing incidence of neoplasms, along with the aging of the population and the associated comorbidities, makes a comprehensive geriatric assessment mandatory to tailor the optimal treatment strategy. The main goal of our study was to validate the application of the G8 score on elderly patients undergoing surgery for kidney cancer, in order to predict the incidence of complications.

The results of the statistical analysis clearly demonstrates how the G8 score was the strongest predictor of overall postoperative complications in an elderly patient with localized renal mass scheduled for surgical treatment with curative intent. Age itself is not closely associated with the onset of complications (*p* = 0.09). It is also important to underline that neither the pathological characteristics of the tumor nor the procedure or surgical approach had a statistically significant impact on complications’ occurrence, as the RENAL score that showed no statistically significant differences between the fragile and non-fragile groups (*p* = 0.73). 

Another interesting finding that emerged from our study is a gender imbalance in the non-frail group patients (45 males vs. 26 females). Smoking habits and sex steroid hormones seem to have a possible role in explaining these gender disparities, but nowadays many studies are investigating the role of biological, genetic, and molecular pathways that explain the gender-related differences in RCC [21,22].

The G8 score is currently used in many clinical settings to determine the elderly’s condition of frailty or not. It was first conceived in 2008, when it emerged that CGA was an excellent tool but not really necessary for all patients, thus introducing the concept of geriatric screening [23]. A study was conducted on 364 patients with a recent diagnosis of cancer and older than 70 years of age, with the aim of testing the G8 as a pre-surgical screening tool to predict the risk of complications. The value of 14 was identified as a cutoff, as it resulted associated to excellent results in terms of both sensitivity and specificity. Following these encouraging results, in-depth studies on G8 application in clinical practice were performed. The score was definitely validated by the same study group in 2011 thanks to the ONCO-DAGE40 project results, a prospective multicenter study which involved 1435 patients in 23 different healthcare facilities [19]. The trial confirmed that the G8 questionnaire was one of the best screening tools available to identify elderly patients with cancer who require a complete and thorough geriatric evaluation, with good sensitivity and an independent prognostic value for one-year survival.

Over time, the G8 score has been studied in various declinations and clinical fields, demonstrating how its use has proven useful and statistically significant [24,25,26,27]. Several studies have been conducted in the surgical setting too, demonstrating the versatility and functionality of the G8 screening tool [28,29,30,31]. In particular, a prospective trial was conducted with the aim to validate G8 as a screening tool in older cancer patients diagnosed with solid neoplasia candidates for surgical treatment, in order to identify those who needed extended CGA. Both CGA and G8 were carried out prior to the surgery. Collected data included thirty-days postoperative complications, postoperative hospital stay, rate of unplanned readmissions, rate of discharges to a rehabilitation unit, and the 1-year mortality. It resulted that G8 is a useful screening tool as patients with an impaired G8 are more at risk of negative postoperative outcomes [32].

Evidence from the literature supporting the use of CGA is very solid, in fact, the International Society of Geriatric Oncology (SIOG) has inserted the CGA in clinical practice with the recent update of the guidelines and the European Organisation for Research and Treatment of Cancer (EORTC), making the G8 screening tool mandatory for all patients aged > 70 years included in the organization’s trials [33,34,35,36]. Concerning urologic neoplasms, the G8 tool has recently been introduced in the EAU guidelines for prostate cancer [37]. 

Treatment options for localized renal cell carcinoma are multiple, ranging from observation alone to radical surgery, and should be tailored according to the patient’s overall health status [38]. Medicine will have to face the global aging of the population, therefore in this context, the G8 screening tool could be able to identify quickly and reliably those in need of a more in-depth screening prior to renal surgery [39]. Some modified versions of the G8 screening have been developed and validated, demonstrating how continuous research is always necessary to further refine this already valid screening tool available to us, making it even more complete and precise [40,41]. This emerging clinical need, along with data that emerged from the above-mentioned studies and the incorporation of G8 in the international guidelines were the main factors that pushed us to the development of our study. 

Moreover, current research aims to compare the use of the G8 with other geriatric assessment scales, with the purpose of consolidating its use. One study compared the G8 score with ECOG. In a cohort of 264 patients, it emerged the G8 score contributes to the prompt identification of patients with poor prognosis and improved the prognostic value of ECOG-PS [42]. A systematic review analyzed the literature on the topic from 2000 to 2017, including nine studies (six prospective and three retrospective, none were randomized controlled trials). It emerged that the variables of the CGA were both prospectively and retrospectively significant predictors of complications of urological surgery. Despite that, the use of CGA is not part of urological daily clinical practice. Therefore, the inclusion of geriatric assessment as part of routine preoperative care in geriatric urology patients should be considered but, further studies are needed because of the lack of randomized controlled trials on preoperative CGAs in urology patients [43]. A recent trial focused on the comparison between G8 to the Charlson Comorbidity Index (CCI) to predict postoperative complications in older patients candidate for major uro-oncologic surgery (radical prostatectomy, partial/radical nephrectomies, radical cystectomies, and nephroureterectomies). It emerged that the complications rate did not correlate with CCI score, while it resulted higher in patients with G8 scores ≤ 14, thus confirming the predictive power of the tool. A further analysis was also performed by dividing patients into three distinct categories (G8 scores of <10, 10–14, and >14), which highlighted different complication rates among the groups. Nevertheless, the subgroup analysis according to kind of surgical procedure did not show significant differences [44]. 

The main limitations of our study were the relatively low sample number of patients and, due to the small size of the population involved in the study, the rare incidence of intraoperative or postoperative complications. A classification of intraoperative complications was not performed, and functional postoperative outcomes were not recorded. Further investigations are needed to confirm the predictive potential of this tool in identifying fragile patients and to explore its additional potential uses, in particular, to predict the degree of functional recovery after surgery.

## 5. Conclusions

Our study demonstrates how the G8 screening tool appears to be an effective and useful instrument for predicting the risk of complications in elderly patients candidates for renal surgery. It is a cost-effective instrument, it only takes a few minutes to be completed and is more reliable than other items in predicting which patients need a more in-depth geriatric and anesthesiologic evaluation.

In a world where the evolution of medicine itself aims at interdisciplinary collaboration and tailored therapy, treatment strategies must be adapted to the clinical conditions and comorbidities of the patient. Taking this into account, an evaluation of the risks/benefits profile of major surgery becomes increasingly necessary, especially considering the global complexity and the needs of geriatric patients. 

G8 configures as a rapid and reliable screening tool in the preoperative evaluation of the patient, and its use should be further extended and improved by also analyzing the post-surgical functional outcome of the elderly patient, providing the clinician with one powerful tool to predict not only complications but also the patient’s chances of recovery in the post-operative period.

## Figures and Tables

**Figure 1 jcm-11-03785-f001:**
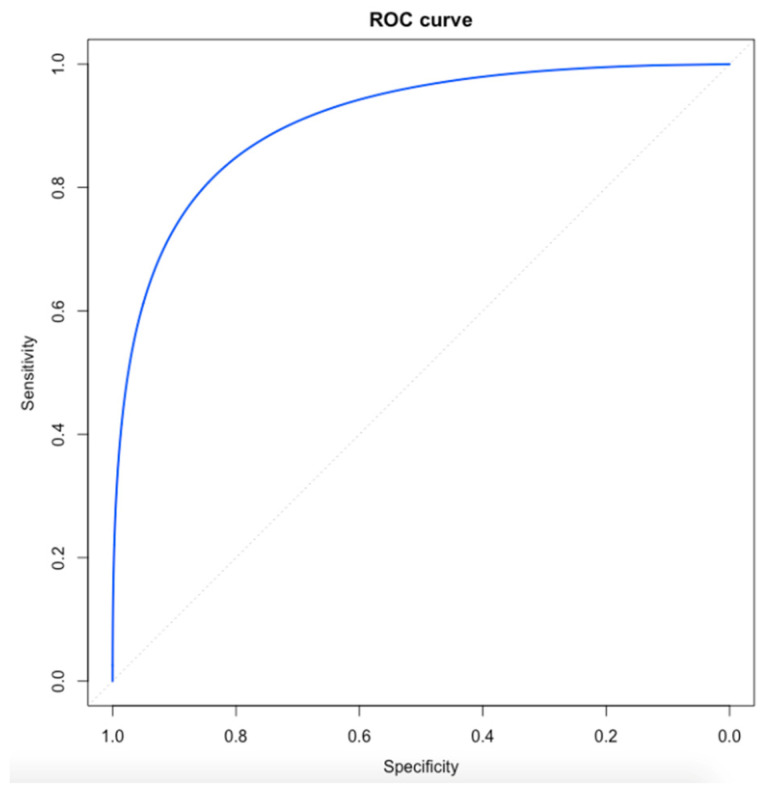
The ROC analysis for the G8 screening tool.

**Table 1 jcm-11-03785-t001:** Descriptive baseline, surgical characteristics, and clinicopathological features for the cohort of 162 patients with renal mass treated with radical or partial nephrectomy according to G8 Screening Tool status.

Variable	Overall	G8 Score ≤ 14	G8 Score > 14	*p*
**Patients**, *n*. (%)	162 (100.0)	91 (56.2)	71 (43.8)	
**Gender**, *n*. (%)				0.13
Male	92 (56.8)	47 (51.6)	45 (63.4)	
Female	70 (43.2)	44 (48.4)	26 (36.6)	
**Age (years)**, median (IQR)	76 (72–80)	77 (73–82)	75 (72–78)	0.16
**BMI (kg/m^2^)**, median (IQR)	25 (23–27)	25 (23–27)	25 (24–27)	0.76
**ECOG**, *n*. (%)				0.04
0	6 (3.7)	5 (5.5)	1 (1.4)	
1	72 (44.4)	32 (35.2)	40 (56.3)	
2	75 (46.3)	48 (52.7)	27 (38.0)	
3	9 (5.6)	6 (6.6)	3 (4.2)	
**CCI**, median (IQR)	3 (2–4)	3 (2–5)	3 (2–3)	0.06
**ASA score**, *n*. (%)				<0.001
≥3	116 (71.6)	51 (56.0)	65 (91.5)	
**Preoperative eGFR (mL/min)**, median (IQR)	56 (43–69)	56 (43–71)	55 (47–67)	0.89
**CKD stage III**, *n*. (%)	73 (45.1)	44 (48.3)	29 (40.8)	0.34
**Preoperative anemia**, *n*. (%)	68 (42.0)	40 (44.0)	28 (39.4)	0.56
**Size of renal mass (mm)**, median (IQR)	47.5 (35–65)	50 (35–63)	46 (35–65)	0.33
**cT**, *n*. (%)				0.41
cT1 (a, b)	122 (75.3)	68 (74.7)	54 (76.1)	
cT2 (a, b)	20 (12.3)	9 (9.9)	11 (15.5)	
cT3 (a, b, c)	15 (9.3)	11 (12.1)	4 (5.6)	
cT4	5 (3.1)	3 (3.3)	2 (2.8)	
**R.E.N.A.L. score**, median (IQR)	8 (6–10)	8 (6–10)	8 (6–10)	0.73
**Surgical technique**, *n*. (%)				0.37
Radical Nephrectomy	94 (58.1)	55 (60.4)	38 (53.5)	
Partial Nephrectomy	68 (41.9)	36 (39.6)	32 (46.5)	
**Surgical approach**, *n*. (%)				0.25
Open	156 (96.3)	89 (97.8)	67 (94.4)	
Minimally-Invasive	6 (7.7)	2 (2.2)	4 (5.6)	
**Type of Ischemia ***, *n*. (%)				0.09
Warm	44 (27.1)	20 (22.0)	24 (33.8)	
Cold	24 (14.9)	16 (28.6)	8 (11.3)	
**Ischemia Time (min.) ***, median (IQR)	17 (15–21)	17 (15–22)	17 (15–21)	0.71
**Blood Loss (mL)**, median (IQR)	300 (200–445)	350 (220–500)	250 (200–400)	0.05
**LND performed**, *n*. (%)	16 (9.9)	11 (12.1)	5 (7.0)	0.28
**Intraoperative complications**, *n*. (%)	43 (26.5)	41 (45.1)	2 (2.8)	<0.001
**Benign Histotype,***n* (%)	14 (8.6)	5 (3)	9 (5.6)	0.58
**pT stage ****, *n*. (%)				0.6
pT1 (a, b)	101 (60.3)	52 (57.1)	49 (69.0)	
pT2 (a, b)	41 (25.3)	25 (27.5)	16 (22.5)	
pT3 (a, b, c)	6 (3.7)	4 (4.4)	2 (2.8)	
**Fuhrman Grade ****, *n*. (%)				0.39
G1–2	89 (55.0)	54 (66.7)	35 (52.2)	
G3–4	59 (36.4)	27 (33.3)	32 (47.8)	
**Overall postoperative complications**, *n*. (%)	56 (34.6)	52 (56.0)	4 (5.6)	<0.001
**Postoperative complications according to Clavien Dindo Classification**, *n*. (%)				<0.001
None	106 (65.4)	38 (42.2)	68 (94.4)	
Grade I	10 (6.2)	8 (8.9)	2 (2.8)	
Grade II	35 (21.6)	33 (36.7)	2 (2.8)	
≥Grade III	11 (6.7)	11 (12.2)	0 (0.0)	
**Length of hospital stay (days)**, median (IQR)	8 (5–11)	9 (7–12)	8 (3–10)	0.07
**30-days readmission**, *n*. (%)	4 (2.5)	1 (1.1)	3 (4.3)	0.21
**Follow-up (months)**, median (IQR)	36 (16–60)	34 (14–58)	36 (17–65)	0.27

* Applied to Partial Nephrectomies. ** Applied to malignant masses. Abbreviations are as follows: IQR: interquartile range; BMI: Body Mass Index; ECOG: Eastern Cooperative Oncology Group; CCI: Charlson Comorbidity Index; ASA: American Society of Anaesthesiologists’ score; eGFR: Estimated glomerular filtration rate; CKD: Chronic Kidney Disease; cT: clinical Tumor stage; R.E.N.A.L.: (R)adius (tumor size as maximal diameter), (E)xophytic/endophytic properties of the tumor, (N)earness of tumor deepest portion to the collecting system or sinus, (A)nterior (a)/posterior (*p*) descriptor and the (L)ocation relative to the polar line; LND: Lymph Node Dissection; pT: pathological Tumor stage.

**Table 2 jcm-11-03785-t002:** Univariate and Multivariate Logistic Regression Analysis of factors significantly associated with perioperative complications according to Clavien-Dindo Classification.

	Univariate			Multivariate		
Variable	OR	95% CI	*p*	OR	95% CI	*p*
**Age** (as cont.)	1.05	0.99–1.12	0.09	1.02	0.93–1.12	0.61
**Sex**						
female	ref.	ref.		ref.	ref.	
male	0.62	0.32–1.19	0.16	0.57	0.26–1.56	0.28
**G8 score**						
>14	ref.	ref.		ref.	ref.	
≤14	21.36	7.98–74.58	<0.001	3.87	11.16–20.41	0.03
**CCI**						
0	ref.	ref.		ref.	ref.	ref.
1	1.42	0.06–15.11	0.78	0.84	0.26–17.33	0.91
≥2	2.36	1.64–14.89	0.04	2.39	1.24–5.15	0.04
**ASA score**						
1, 2	ref.	ref.		ref.	ref.	
≥3	0.95	0.47–1.98	0.88	0.81	0.27–2.32	0.69
**Preop. Anemia**						
no	ref.	ref.		ref.	ref.	
yes	1.80	0.94–3.51	0.08	1.87	0.68–5.51	0.24
**Preop. eGFR** (as cont.)	0.99	0.97–1.02	0.74	1.01	0.97–1.03	0.89
**R.E.N.A.L.**	0.98	0.86–1.13	0.83	0.88	0.67–1.14	0.33
**cT**						
T1 (a, b)	ref	ref.		ref.	ref.	
T2 (a, b)	0.46	0.13–1.45	0.19	0.34	0.61–1.69	0.19
T3 (a, b, c)	0.92	0.27–2.75	0.88	0.42	0.68–2.47	0.64
T4	2.76	0.44–21.51	0.27	2.06	0.71–15.6	0.32
**Surgical Procedure**						
Radical Nephrectomy	ref.	ref.		ref.	ref.	
Partial Nephrectomy	0.67	0.34–1.31	0.25	0.89	0.31–2.59	0.38
**Surgical Technique**						
Open	ref.	ref.		ref.	ref.	
Minimally-Invasive	0.97	0.13–5.14	0.97	2.12	0.21–3.53	0.83

Abbreviations are as follows: OR: Odds Ratio; CI: Confidence Interval; CCI: Charlson Comorbidity Index; ASA: American Society of Anaesthesiologists’ score; eGFR: Estimated glomerular filtration rate; R.E.N.A.L.: (R)adius (tumor size as maximal diameter), (E)xophytic/endophytic properties of the tumor, (N)earness of tumor deepest portion to the collecting system or sinus, (A)nterior (a)/posterior (*p*) descriptor and the (L)ocation relative to the polar line; cT: clinical Tumor stage.

## Data Availability

Data are anonymized and storage at Urology clinic in the University of Trieste at the Department of Medical, Surgical and Health Sciences.

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
