# Peer review of "Reliable Prediction of Post-Operative Complications’ Rate Using the G8 Screening Tool: A Prospective Study on Elderly Patients Undergoing Surgery for Kidney Cancer"

_jcm, 2022, doi:10.3390/jcm11133785_

Round 1
Reviewer 1 Report
The study is interesting, but it needs some improvements before being considered suitable for publication in JCM. The improvements needed are listed in detail below.
TITLE: The study needs a more attention-catching title, and also one more adherent to the data presented by the Authors. For example: " RELIABLE PREDICTION OF POST-OPERATIVE COMPLICATION RATE USING THE G8 SCREENING TOOL: A PROSPECTIVE STUDY ON ELDERLY PATIENTS UNDERGOING SURGERY FOR KIDNEY CANCER"
ABSTRACT: "The G8 screening tool is a powerful tool able to identify the risk profile pf elderly patients, in order to...." .
Do not talk about prognosis in the abstract, prognostic evaluation is not the focus of this paper.
Do not mention the cut off (14) in the abstract, readers cannot understand this concept by reading the abstract alone . The abstract should be self-standing.
...developed post-operative complications of any kind.. Of these, 11 were major complications.....
"Our results suggest that the G8 Screening Tool could be an effective and useful instrument to predict the risk of overall complications in elderly patients prior to renal surgery ".
This concept, that is the conclusion of the Abstract, is a bit like the central concept of this study. So you should expand this concept in the paper Introduction and the in the Discussion, and speculate on the possible clinical relevance of the G8 Screening Tool. What is it, other than knowing in advance which patients are more at risk of complications? Which actions could be undertaken, on the basis of the G8 Screening Tools results, prior to surgery, in the frail group of patients? Particularly regarding the risk major complications, should the "frail" patients be managed differently? Should the anesthesiologists be involved upfront? What do the Authors suggest to the readers, in order to convince them about the clinical relevance of the G8 Screening Tool?
INTRODUCTION, line 37-38: are these lines necessary? Or should they be omitted?.
line 40: ...the oncological management in the elderly is a real challenge, especially in case of localised kidney neoplasms
line 45-53: re-phrase, unclear
line 57: The primary endpoint of our study is to determine the role of the G8 score in predicting intra and postoperative complications in cases going to be submitted to renal surgery.
Materials and Methods: line 94: Subtitle: The G8 Screening Tool The G8 Screening Tool includes seven items.....
line 102: The role...
line 108: ...geriatric risk profile based on the G8...
Results: line 129: 41 patients, identified as frail by the G8 score, versus 2 patients in the non-frail group......
line 133: why do eyouneed to state the follow up time in the results section? It is not not part of the focus of this paper and can be omitted.
TABLE2: A striking gender imbalance can be noted in the non-frail group of patients (45 males, 26 females). The Authors should comment on this point, here and in the Discussion. Any gender-related clinical correlation possible? Why do the Authors think there are less females in the G8 score>14 group? A couple of citations could be added here, regarding gender-related differences in kidney cancer, a very novel and fashionable topic: Mancini M., at. al, IJMS 2020, and Ishiyama Y, at al., Curr Oncol 2021. A comment on gender-related data would make the paper much more updated and interesting for the readers of JCM.
Results, line 158: None of the single predictors...
line 164: that the G8 is a reliable instrument to predict...
Discussion: line 179-189: re-phrase, unreadable. Do not complement yourself in the Discussion, just underline your data and make a comparison to previously published evidence.
line 200: "The G8 questionnaire, which takes a few minutes to be completed, is currently one of the screening tools available to identify elderly patients with cancer who require a more complete and thorough geriatric and anesthesiological evaluation". THIS IS THE CENTRAL CONCEPT OF THIS PAPER and it should be clearly stated and repeated in the Conclusions section as well.
line 205: the G8 screening tool...
line 208-211: No need to write the title of the citation, just put the citation number at the end of the sentence.
line 212: the use of the CGA is very solid...
line 235-237: The G8 Screening Tool could be able to quickly and reliably identify patients in need of a more in depth screening prior to renal surgery.
line 241-246: re-phrase the Limitations of the study in a more readable way
Conclusions: Our study aligns with existing literature on this topic....Specify, rather than stating that your study is like the others, which are the new concepts of your own work and which gaps does it fills. Uderline the strengths of your data and the original parts of your results.
line 250: Do not mention the functional results because they are not the focus of this paper.
line 252: The G8 Screening Tool seems to be a rapid, functional and effective tool.... Its use should be further extended to...EXPLAIN MORE CLEARLY, reconnect with the Introduction and the Title and wrap up your paper with a clear and clinically relevant statement.
Author Response
- The study needs a more attention-catching title, and also one more adherent to the data presented by the Authors.
R:We thank the reviewer for this excellent advise. Accordingly, we updated the title to “Reliable prediction of post-operative complications’ rate using the G8 screening tool: a prospective study on elderly patients undergoing surgery for kidney cancer”.
- "The G8 screening tool is a powerful tool able to identify the risk profile pf elderly patients, in order to...." . Do not talk about prognosis in the abstract, prognostic evaluation is not the focus of this paper. Do not mention the cut off (14) in the abstract, readers cannot understand this concept by reading the abstract alone. The abstract should be self-standing. ...developed post-operative complications of any kind.. Of these, 11 were major complications..... "Our results suggest that the G8 Screening Tool could be an effective and useful instrument to predict the risk of overall complications in elderly patients prior to renal surgery ".
R:We thank the reviewer for this remarkable comment. Accordingly, we updated the abstract (see lines 17, 22, 26), particurly omitting sentences about prognosis and cut-off in the abstract
- line 37-38: are these lines necessary? Or should they be omitted? line 40: ...the oncological management in the elderly is a real challenge, especially in case of localised kidney neoplasms. line 45-53: re-phrase, unclear. line 57: The primary endpoint of our study is to determine the role of the G8 score in predicting intra and postoperative complications in cases going to be submitted to renal surgery.
R:We thank the reviewer for this remarkable comment. Accordingly, we updated the manuscript (Omitted line 37-38, stylistic and conceptual revision as suggested (Line 45-53 /line 56-57)
- line 94: Subtitle: The G8 Screening Tool includes seven items. line 102: The role...line 108: ...geriatric risk profile based on the G8... 129: 41 patients, identified as frail by the G8 score, versus 2 patients in the non-frail group......line 133: why do eyouneed to state the follow up time in the results section? It is not not part of the focus of this paper and can be omitted.
- TABLE2: A striking gender imbalance can be noted in the non-frail group of patients (45 males, 26 females). The Authors should comment on this point, here and in the Discussion. Any gender-related clinical correlation possible? Why do the Authors think there are less females in the G8 score>14 group? A couple of citations could be added here, regarding gender-related differences in kidney cancer, a very novel and fashionable topic: Mancini M., at. al, IJMS 2020, and Ishiyama Y, at al., Curr Oncol 2021. A comment on gender-related data would make the paper much more updated and interesting for the readers of JCM.
R:We thank the reviewer for this important comment. Accordingly we updated the manuscript (line 94, 105,108) - omitted the state on follow up time – Inserted a line (line 137) on striking gender imbalance in the non-frail group of patients ) and added 2 citations on the theme in th discussion (line 2017: citation 21. Mancini, M.; Righetto, M.; Baggio, G. Gender-Related Approach to Kidney Cancer Management: Moving Forward. Int. J. Mol. Sci. 2020, 21, 3378, doi:10.3390/ijms21093378., 22. Peired, A.J.; Campi, R.; Angelotti, M.L.; Antonelli, G.; Conte, C.; Lazzeri, E.; Becherucci, F.; Calistri, L.; Serni, S.; Romagnani, P. Sex and Gender Differences in Kidney Cancer: Clinical and Experimental Evidence. Cancers 2021, 13, 4588, doi:10.3390/cancers13184588.
- line 158: None of the single predictors.. line 164: that the G8 is a reliable instrument to predict...
R:We thank the reviewer for this important comment. Accordingly, we updated the manuscript (line 158, 164)
- line 179-189: re-phrase, unreadable. Do not complement yourself in the Discussion, just underline your data and make a comparison to previously published evidence. line 200: "The G8 questionnaire, which takes a few minutes to be completed, is currently one of the screening tools available to identify elderly patients with cancer who require a more complete and thorough geriatric and anesthesiological evaluation". THIS IS THE CENTRAL CONCEPT OF THIS PAPER and it should be clearly stated and repeated in the Conclusions section as well. line 205: the G8 screening tool...line 208-211: No need to write the title of the citation, just put the citation number at the end of the sentence. line 212: the use of the CGA is very solid...line 235-237: The G8 Screening Tool could be able to quickly and reliably identify patients in need of a more in depth screening prior to renal surgery. line 241-246: re-phrase the Limitations of the study in a more readable way
R:We thank the reviewer for this important comment. Accordingly, we updated the manuscript drastically reviewed adding a paragraph on striking gender imbalance in the non-frail group of patients (line 203), focused the attention on the centrality of G8 and its practical usability. (line 209-223-234). Added a citation that compares G8 score to CGA and CCI in some recent articles like suggested (citation 32 and 44 L251 and 262)
- Conclusions: Our study aligns with existing literature on this topic....Specify, rather than stating that your study is like the others, which are the new concepts of your own work and which gaps does it fills. Uderline the strengths of your data and the original parts of your results. line 250: Do not mention the functional results because they are not the focus of this paper. line 252: The G8 Screening Tool seems to be a rapid, functional and effective tool.... Its use should be further extended to...EXPLAIN MORE CLEARLY, reconnect with the Introduction and the Title and wrap up your paper with a clear and clinically relevant statement.
R:We thank the reviewer for this important comment. Accordingly, we updated the manuscript drastically reviewed adding a paragraph on possible use of G8 function evaluation in the future (line 280, line 291)
Reviewer 2 Report
This article is an interesting study regarding the G8 screening tool in renal tumor patients. The clinical information were precisely collected and the article is well-written.
First, it seems this study would be given as a presentation in 2018 (MP59-03 https://www.auajournals.org/doi/pdf/10.1016/j.juro.2018.02.1854). The study population and analyses are same. However, the study duration looks different. Please clarify this discrepancy.
Abstract:
P1L26 p<0.001 in Table3?
Introduction
P1L36 References looks old. Please modify appropriately. For example, as for EAU guidelines on Renal Cell Carcinoma(ref#1), there are another latest article. ref#2 could be an article from AUA guidelines.
P2L54 Please add a new reference.
Materials and Methods:
P2L64 Please clarify the discrepancy regarding the study duration between your article and presentation.
P2L69 In introduction, 65 y.o. was used as an example for elder patients. Please clarify the reason you used patients larger than or equal to 70 years.
P2L73-77 Please add appropriate references for scores.
P2L90 Please clarify the two GU pathologist at each center in your article or acknowledgement.
P2L94 Please add a reference for the G8 screening score to the M&M section.
P3L102 and 118 endpoints are unclear. What kinds of disease or status are included in the perioperative complications as endpoints? Please describe more precisely.
P4 Table2 The number with pathological results for malignancy was 148. The remaining 14 patients were benign? Benign vs Malignant can be related to the endpoints? Please clarify this in M&M section. Furthermore, Overall and dedicated postoperative complications are unclear. Please clarify.
Results:
P6L164 the G8 is an extremly useful instrument to predict... Your impression should not be in the Result section.
P6L164 p<0.001 in Table3?
P6 Table 3 BMI can be related to G8 screening tool. clinical size can be related to R.E.N.A.L score. Please remove BMI and clinical size from the analyses. Additionally, as for R.E.N.A.L score, please show the variable as categorical, not continuous because other scores were shown as categorical.
Discussion:
P7L190 It would be better to show the description regarding the G8 screening tool in the introduction section.
P8L219 Currently,,, Is there other latest articles?

Author Response
- First, it seems this study would be given as a presentation in 2018 (MP59-03 https://www.auajournals.org/doi/pdf/10.1016/j.juro.2018.02.1854). The study population and analyses are same. However, the study duration looks different. Please clarify this discrepancy
R:We thank the reviewer for this excellent comment. We acknowledged the time frame incorrectly reported in the abstract. We confirm that the study was prospectively carried out between 2015 - 2019.
- P1L36 References looks old. Please modify appropriately. For example, as for EAU guidelines on Renal Cell Carcinoma(ref#1), there are another latest article. ref#2 could be an article from AUA guidelines. P2L54 Please add a new reference.
R:We thank the reviewer for this important comment. Accordingly we updated the manuscript (P1L36 - Updated the citation on EAU and AUA Guidelines ). L54 added the reference: “9. Hamaker, M.E.; Jonker, J.M.; de Rooij, S.E.; Vos, A.G.; Smorenburg, C.H.; van Munster, B.C. Frailty Screening Methods for Predicting Outcome of a Comprehensive Geriatric Assessment in Elderly Patients with Cancer: A Systematic Review. Lancet Oncol. 13, e437–e444, doi:10.1016/S1470-2045(12)70259-0.”
- P2L69 In introduction, 65 y.o. was used as an example for elder patients. Please clarify the reason you used patients larger than or equal to 70 years.
R:We thank the reviewer for this important comment. Accordingly we updated the manuscript (P2L73 – Added the reference 10: 10. Orimo, H.; Ito, H.; Suzuki, T.; Araki, A.; Hosoi, T.; Sawabe, M. Reviewing the Definition of “Elderly.” Geriatr. Gerontol. Int. 2006, 6, 149–158, doi:10.1111/j.1447-0594.2006.00341.x. )
- P2L73-77 Please add appropriate references for scores.
R:We thank the reviewer for this important comment. Accordingly we updated the manuscript (P2L76- L81 – Added the references on the scores)
- P2L94 Please add a reference for the G8 screening score to the M&M section.
R:We thank the reviewer for this important comment. Accordingly we updated the manuscript adding a citation (L100, citation 19: 19. Soubeyran, P.; Bellera, C.; Goyard, J.; Heitz, D.; Cure, H.; Rousselot, H.; Albrand, G.; Servent, V.; Saint Jean, O.; Roy, C.; et al. Validation of the G8 Screening Tool in Geriatric Oncology: The ONCODAGE Project. J. Clin. Oncol. 2011, 29, 9001–9001, doi:10.1200/jco.2011.29.15_suppl.9001.
- P3L102 and 118 endpoints are unclear. What kinds of disease or status are included in the perioperative complications as endpoints? Please describe more precisely.
R:We thank the reviewer for this important comment. Accordingly we updated the manuscript (P3L106-110)
- P4 Table2 The number with pathological results for malignancy was 148. The remaining 14 patients were benign? Benign vs Malignant can be related to the endpoints? Please clarify this in M&M section. Furthermore, Overall and dedicated postoperative complications are unclear. Please clarify.
R:We thank the reviewer for this important comment. Accordingly we updated the manuscript adding a line (P3L143) added the 14 benign histotypes in table 2(P4).
- P6L164 the G8 is an extremly useful instrument to predict... Your impression should not be in the Result section.
- P6 Table 3 BMI can be related to G8 screening tool. clinical size can be related to R.E.N.A.L score. Please remove BMI and clinical size from the analyses. Additionally, as for R.E.N.A.L score, please show the variable as categorical, not continuous because other scores were shown as categorical.
R:We thank the reviewer for this important comment. Accordingly, we updated the manuscript editing the P6Table3: eliminated BMI and clinical size. Renal score stated as categorical and removed personal impression (P6L175)
- P8L219 Currently,,, Is there other latest articles?
R:We thank the reviewer for this important comment. Accordingly, we updated the manuscript drastically reviewed adding a paragraph on striking gender imbalance in the non-frail group of patients (line 203), focused the attention on the centrality of G8 and its practical usability. (line 209-223-234). Added a citation that compares G8 score to CGA and CCI in some recent articles like suggested (citation 32 and 44 L251 and 262).
Thank you for your observations. We hope that our revisions will improve the manuscript.
Round 2
Reviewer 1 Report
The paper has been significantly improved both conceptually and grammatically. I find it now suitable for publication.
Reviewer 2 Report
Well revised and looks nice.